# Physical Exercise and the Gut Microbiome: A Bidirectional Relationship Influencing Health and Performance

**DOI:** 10.3390/nu16213663

**Published:** 2024-10-28

**Authors:** Sanish Varghese, Shrinidhi Rao, Aadam Khattak, Fahad Zamir, Ali Chaari

**Affiliations:** Department of Biochemistry, Premedical Division, Weill Cornell Medicine–Qatar, Qatar Foundation, Education City, Doha P.O. Box 24144, Qatar; ssv4001@qatar-med.cornell.edu (S.V.); smr4004@qatar-med.cornell.edu (S.R.); amk4018@qatar-med.cornell.edu (A.K.); faz4003@qatar-med.cornell.edu (F.Z.)

**Keywords:** physical activity, exercise, training, athletes, gut microbiota, probiotics

## Abstract

**Background/Objectives:** The human gut microbiome is a complex ecosystem of microorganisms that can influence our health and exercise habits. On the other hand, physical exercise can also impact our microbiome, affecting our health. Our narrative review examines the bidirectional relationship between physical activity and the gut microbiome, as well as the potential for targeted probiotic regimens to enhance sports performance. **Methods:** We conducted a comprehensive literature review to select articles published up till January 2024 on the topics of physical exercise, sports, probiotics, and gut microbiota from major scientific databases, incorporating over 100 studies. **Results:** We found that the impact of physical activity on the gut microbiome varies with the type and intensity of exercise. Moderate exercise promotes a healthy immune system, while high-intensity exercise for a long duration can cause a leaky gut and consequent systemic inflammation, which may disrupt the microbial balance. Combining aerobic and resistance training significantly affects bacterial diversity, linked to a lower prevalence of chronic metabolic disorders. Furthermore, exercise enhances gut microbiome diversity, increases SCFA production, improves nutrient utilization, and modulates neural and hormonal pathways, improving gut barrier integrity. Our findings also showed probiotic supplementation is associated with decreased inflammation, enhanced sports performance, and fewer gastrointestinal disturbances, suggesting that the relationship between the gut microbiome and physical activity is mutually influential. **Conclusions:** The bidirectional relationship between physical activity and the gut microbiome is exemplified by how exercise can promote beneficial bacteria while a healthy gut microbiome can potentially enhance exercise ability through various mechanisms. These findings underscore the importance of adding potential tailored exercise regimens and probiotic supplementation that consider individual microbiome profiles into exercise programs.

## 1. Introduction

The human gut microbiome is a complex community of microorganisms, including bacteria, archaea, fungi, viruses, and protozoa. It comprises over a thousand bacterial species that can significantly influence the host’s overall well-being [1,2]. In a healthy gut, anaerobic species are typically responsible for producing anti-inflammatory short-chain fatty acids (SCFAs) through the fermentation of complex dietary sugars [3]. SCFAs play crucial roles in maintaining gut integrity and modulating the immune system. For example, SCFA butyrate serves as the primary energy source for colonocytes, enhancing epithelial barrier function and preventing the translocation of pathogens and toxins into the bloodstream [4]. Additionally, SCFAs can influence the immune system by promoting the differentiation of regulatory T cells, which help to maintain immune tolerance and reduce inflammation [5]. However, the beneficial metabolic activities of these anaerobic species are often compromised in conditions of gut dysbiosis, leading to a reduced production of SCFAs and other byproducts [3].

The delicate balance of microbial species and their byproducts is intricately linked to overall health, influencing a range of diseases. Extensive research has revealed significant associations between microbial imbalances and chronic conditions such as metabolic syndrome, obesity, antibiotic use, and depression [6,7]. These conditions are often characterized by a shift from a dominance of anaerobes to facultative species, which can disrupt normal metabolic processes and lead to systemic inflammation [3]. Such microbial imbalances reduce microbial diversity, which can adversely affect the gut’s functionality in multiple ways. For instance, a less diverse microbiome diminishes the gut’s efficiency in metabolizing nutrients, which can contribute to the energy imbalance observed in obesity and metabolic syndrome. A diverse microbiome enhances the capacity for polysaccharide fermentation, leading to increased SCFA production, which not only provides energy to the host but also supports glucose homeostasis [8]. A decreased gut microbiota diversity has been linked to a higher prevalence of coronary vascular disease and nonalcoholic fatty liver disease [9]. Moreover, reduced microbial diversity also weakens the gut’s defensive mechanisms against pathogenic bacteria such as *Proteobacteria*, further exacerbating the risk of dysbiosis and its associated health conditions [10,11].

This interplay between microbial diversity and metabolic health is intricately connected to physical activity. Regular exercise has been shown to promote microbial diversity, facilitating a more favorable environment for beneficial microbial populations that enhance SCFA production. The effects of physical activity on overall well-being include improvements in physical, mental, and gut health, achieved through complex biological mechanisms and interactions. Regular engagement in physical exercises not only helps to manage and reduce the risk of developing metabolic syndrome but also improves mental health, anxiety, and depression [12,13]. Exercise regulates the immune system by prompting an acute rise in interleukin-6 (IL-6), which helps to reduce inflammation [14]. However, extended durations of strenuous exercise can lead to a transient increase in inflammation due to muscle tissue damage [14]. These interactions also extend to gut health, where regular exercise is correlated with the mitigation of disease states associated with gut dysbiosis [15,16,17]. The exact causal mechanisms behind these correlations are still under investigation. Recent studies reveal that while high-intensity, long-duration exercise can cause a leaky gut and dysbiosis, low-to-moderate exercise maintains microbiome health [18]. The effects of physical activity on the microbiome vary significantly with the type of exercise, such as endurance training, resistance training, and different sports [19].

Research demonstrates that athletes often develop distinct microbiome profiles after exercise, marked by increased beneficial microbial species and SCFAs [20]. Additionally, higher muscle strength in older adults is associated with specific microbiome compositions, indicating that a healthy microbiome can contribute to maintaining physical strength and functionality as we age [21]. Conversely, a compromised gut barrier can lead to joint inflammation, adversely affecting movement and physical performance [22].

Despite the emerging evidence of the relationship between physical activity and gut microbiome health, several gaps remain in the current literature. While the fact that exercise influences microbial diversity and SCFA production is documented, understanding specific bidirectional relationships between gut microbiome profiles and exercises of various intensities requires a comprehensive exploration. Our review aims to integrate findings from animal and human studies to consolidate current knowledge, identify gaps, and direct future research toward the relationship between physical activity and gut microbiota.

## 2. Methods: Search Strategy and Selection Criteria

We conducted a comprehensive literature review in January 2024 to select articles on physical exercise, sports, and gut microbiota from PubMed, Google Scholar, Scopus, and NCBI. The search included keywords such as “physical activity”, “physical training”, “aerobic exercise”, “high-intensity interval training (HIIT)”, “resistance training”, “athletic performance”, “elite athletes”, “exercise intensity”, “gut microbiota”, “microbiota diversity”, “probiotics”, “SCFA”, and “metabolic health”. We screened article abstracts to assess whether they should be included in the full-text screening. We included over 100 animal and human original research studies that specifically examined the relationship between gut microbiota and physical exercise. Non-peer-reviewed articles and those on relevant topics that do not directly address the review’s focus were excluded. Our narrative review underscores the importance of physical exercise, its connection with gut microbiota, and its beneficial effects on chronic illnesses and athletic performance. The collected data encompass various parameters, including the type of exercise, shifts in bacterial species composition, dosage and strains of administered probiotics, research design, study duration, geographic location, sample size, participants’ age, and participants’ Body Mass Index (BMI).

## 3. Gut Microbiome Effects of Exercise in Animals

Exercise is increasingly recognized as influencing the gut microbiome, impacting its composition and function [23]. Various types of physical activity, such as aerobic exercise and resistance training, can alter the diversity and metabolic activity of gut microbes. This chapter explores the relationship between physical activity and the gut microbiome in animal models (Appendix A), providing a comprehensive overview of how various forms of exercise influence the microbial communities within the gut.

### 3.1. Voluntary Aerobic Exercise

Voluntary aerobic exercise is increasingly recognized for its influence on gut health. In a laboratory study by Mika et al., rats housed in cages equipped with running wheels were allowed voluntary wheel access to perform voluntary aerobic exercise during the experimental period [24,25]. Compared to rats who were assigned to the sedentary (control) group, the voluntary aerobic exercise group showed an increased abundance of beneficial bacterial genera like *Lactobacillus* and *Bifidobacterium* [25]. *Lactobacillus* is known for its potential to enhance human intestinal barrier integrity. It plays a crucial role in preventing and treating gastrointestinal infections, reducing intestinal inflammation, improving digestive processes, and enhancing nutrient absorption [26]. On the other hand, *Bifidobacteria* has anti-inflammatory and antiviral properties. It aids in regulating the immune system and improves gut health by assisting in the assimilation of dietary fibers and regulating fat storage [27]. The impact of voluntary aerobic exercise can extend to increasing the abundance of other beneficial bacterial families such as *Ruminococcaceae* and *Lachnospiraceae* [24]. These changes in the microbiome also lead to a higher overall concentration of n-butyrate, an SCFA that confers many benefits for gastrointestinal health and immune system health [28]. Overall, the evidence suggests that voluntary aerobic exercise is related to an increase in the abundance of numerous genera and families that have beneficial effects in enhancing gut health and promoting the growth of healthy bacteria in the gut.

### 3.2. Moderate Aerobic Exercise

Building on the effects observed with voluntary aerobic exercise, moderate aerobic exercise seems to have discernible effects on the gut microbiome of mice. In Lamoureux, Grandy, and Langille’s controlled laboratory study, 42 mice were split into three groups: a voluntary exercise group (10 mice), a forced exercise group (11 mice), and a sedentary control group (21 mice). The intervention (exercise) was implemented over 8 weeks, and the gut microbial diversity, changes in inflammatory markers, and lean body mass were measured. The authors had not observed any significant changes in the overall gut microbial diversity or inflammatory markers after mice underwent moderate aerobic exercise; however, advanced computational methods such as random forest machine learning were able to discern notable differences between the gut microbiome of mice that moderately exercised and mice that did not [29]. The study also found increases in the *Bacteroides* and *Lactobacillus* genus. Both voluntary and moderate aerobic exercise increased certain useful genera in the gut microbiome, showing its beneficial effects on the gut microbiome; however, moderate aerobic exercise did not see as much change in gut microbial diversity.

### 3.3. Intense Aerobic Exercise

In contrast to moderate aerobic exercise, intense aerobic exercise enhances the growth of additional genera such as *Firmicutes* and *Bacteroidetes,* which in turn promote positive health output. Research on thoroughbred racehorses has shown that intense aerobic exercise was associated with significant shifts in the levels of the *Bacteroidetes* and *Firmicutes* phyla [30]. Both of the phyla increased in abundance. It is noteworthy as both of these phyla are beneficial individually, but an imbalance in their ratios (more *Firmicutes* as compared to *Bacteroidetes*) can cause significant physiological harm [31]. The *Firmicutes* phylum plays an important role in digestive health and energy harvesting [32]. Additionally, the *Bacteroidetes* phylum is involved in the breakdown of complex carbohydrates, a process that requires significant aid from bacteria [33], helping in better absorption and utilization. A recurring theme across all forms of aerobic exercise is the increase in beneficial bacterial strains aiding in better gut health. Voluntary aerobic exercise saw an increase in the abundance of numerous genera and families. Moderate aerobic exercise did not see as much change in gut microbial diversity but did see a change in some gut microbial species. Intense exercise resulted in alterations in the two most populous good genera of the gut microbiome, *Bacteroidetes* and *Firmicutes*.

### 3.4. Resistance Training

Resistance training is a type of exercise that involves performing physical activities to increase strength. Research has shown that resistance training may confer benefits to the gut microbiome in animal models. Research on rats has shown that resistance training altered the gut microbiota composition, with an increase in *Lactobacillus* and *Bifidobacterium* [34]. Resistance training protects against autoimmune diseases. This protection is achieved by changing the composition or function of the gut microbiota, particularly *Akkermansia muciniphila*, which in turn leads to a reduction in the activity or presence of T helper 17 (Th17) cells [35]. Similar to voluntary aerobic exercise, studies on resistance training suggest that it is good for the gut microbiota and overall gut health by conferring an increase in the abundance of the *Lactobacillus* and *Bifidobacterium* genera.

## 4. Effects of Different Types of Exercise on the Human Gut Microbiome

Research consistently supports that those various types of exercise influence human gut microbiome composition in ways similar to those observed in animal models (Appendix A, Figure 1). Interventions utilizing multiple exercise routines (aerobic and resistance) had more of a significant influence on bacterial diversity than, for instance, resistance training alone. Studies indicate that low-intensity exercise programs may have a limited positive impact on gut microbiota diversity compared to high-intensity exercise. However, research shows that intense, prolonged exercise could lead to an increase in inflammation [18,36]. Regardless of the type of exercise intervention, most studies in the tables reveal a similar trend of observations concerning the effects on the major phyla of the gut microbiome as explored in this chapter.

### 4.1. Overall Gut Microbiota Diversity

Exercise has repeatedly been demonstrated to positively influence the alpha diversity of the gut microbiota, contributing to overall gut health. For instance, swimmers who participated in a 7-week high-intensity interval training (HIIT) program exhibited a significant increase in alpha diversity within their fecal microbiota [37]. Similarly, a 6-week endurance training was associated with increased beta diversity in a cohort of previously sedentary overweight women [38]. Notably, combining multiple exercise modalities, such as different aerobic exercises and resistance training, appears to exert a more pronounced effect on bacterial diversity compared to employing just one type of exercise. However, intense, prolonged physical activities could lead to an increase in inflammation, suggesting a delicate balance between exercise intensity and its health benefits [18,36]. Intense physical activity can also increase appetite and lead to higher energy intake, often driving athletes to consume energy-dense diets. These diets, particularly those high in fat and refined carbohydrates, can negatively alter the gut microbiome by promoting the growth of pro-inflammatory bacteria and reducing microbial diversity [39]. Thus, it can be concluded that exercise is associated with human gut microbiome diversity, which can be beneficial for overall health.

### 4.2. The Role of the Firmicutes Phylum in Gut Health

Engaging in regular physical activity is correlated with an increase in the bacteria of the dominant phylum *Firmicutes*, which plays a vital role in various gut functions and overall intestinal health [40]. Notably, individuals who engage in regular physical exercise show a marked increase in the abundance of SCFAs produced by *Firmicutes*, particularly those from the *Ruminococcaceae*, *Lachnospiraceae*, and *Erysipelotrichaceae* families [41,42,43]. *Ruminococcus* is a diverse genus of bacteria within the gut microbiome, encompassing both beneficial and potentially harmful species [44]. These bacteria have been found to respond positively to increases in physical activity, with their populations rising as a person becomes more physically active [43,45,46,47,48,49,50,51,52]. *Ruminococcus* includes two species, *Ruminococcus gauvreaui* and *Faecalibacterium prausnitzii*, both of which are associated with enhanced health outcomes. Higher levels of *Ruminococcus gauvreaui* correlate with improved cardiorespiratory fitness and greater insulin sensitivity [44]. *Faecalibacterium prausnitzii* produces the SCFA succinate, which is a substrate for intestinal gluconeogenesis, a process that improves glucose homeostasis [53]. Research has found that *Faecalibacterium prausnitzii* levels are significantly higher in women who engaged in high physical activity in the previous week [54]. Additionally, endurance runners were found to have higher levels of *Faecalibacterium prausnitzii* and SCFA succinate compared to healthy non-athletes [51].

The *Lachnospiraceae* family includes SCFA-producing bacteria such as *Dorea*, *Coprococcus*, and *Roseburia* [55]. *Dorea* produces SCFA acetate, a compound directly linked to glucose levels and known for its positive influence on glucose metabolism [56]. Research has shown that the abundance of *Dorea* is higher in individuals who engage in physical activity, suggesting that regular exercise may amplify its beneficial effects [38,49,57]. *Coprococcus* produces SCFA butyrate, which reduces the severity of atopic conditions such as atopic dermatitis [58]. *Coprococcus* is significantly enriched in individuals who engage in regular physical activity [42,47,49,59,60]. *Roseburia* is another genus of bacteria that produces SCFA butyrate. The abundance of *Roseburia* is significantly higher in actively exercising adults, demonstrating a beneficial impact on gut metabolism and immunity through butyrate production. Notably, *Dorea, Coprococcus*, and *Roseburia* have also been found to increase following a 21.1 km athletic marathon or a 3748.91 km rowing event lasting 33 days and 22 h [47,57].

Other *Firmicutes* have been associated with enhanced gut health and improved metabolic responses, including species like *Streptococcus*, *Romboutsia*, *Holdemanella*, *Ruminococcaceae*, *Blautia*, *Ruminiclostridium*, *Clostridium phoceensis*, *Streptococcus suis*, *Clostridium bolteae*, *Lactobacillus*, *Anaerostipes hadrus*, and *Veillonella*. Studies have shown that *Streptococcus* and *Romboutsia* are more abundant in physically active adults, suggesting a link between these bacteria and regular physical activity [61,62,63]. Notably, specific *Streptococcus* species have demonstrated anti-inflammatory properties in vitro by significantly reducing cytokines such as TNF-α, IL-1β, and IL-6 [47,50,52].

*Holdemanella* levels have been observed to increase dramatically following a weeks-long exercise intervention [64]. Research has shown that *Holdemanella* can mitigate hyperglycemia by restoring levels of the hormone GLP-1, which is crucial for blood sugar regulation [65]. Studies involving athletes have revealed increases in *Ruminococcaceae*, *Blautia*, *Ruminiclostridium*, and *Clostridium phoceensis* after moderate-intensity exercises, such as treadmill workouts [50].

Further studies focusing on elite athletes, including those from 16 different sports, have indicated a considerable predominance of *Firmicutes* members like *Streptococcus suis*, *Clostridium bolteae*, *Lactobacillus*, and *Anaerostipes hadrus* in those participating in moderately active sports, such as fencing [62]. High-intensity sports participants, such as field hockey and rowing athletes, show higher levels of *Lactobacillus acidophilus* and *Faecalibacterium prausnitzii* [62].

Observational and interventional studies have consistently shown that physical activity increases SCFA-producing bacteria like *Veillonella* [45,63,66,67,68]. This is particularly noteworthy as *Veillonella,* a lactate-utilizing species when colonized in mice, has been shown to improve exercise performance. Additionally, *Veillonella* is abundantly found in the gut microbiomes of elite athletes post-marathon, highlighting its potential benefits to athletic performance [69].

These findings underscore the profound impact of physical activity on the gut microbiome. Maintaining an active lifestyle can significantly alter and potentially enhance the growth of *Firmicutes*, which are associated with promoting better health and athletic performance.

### 4.3. The Role of the Bacteroidetes Phylum in Gut Health

*Bacteroidetes* are a group of gut commensals that play a crucial role in protecting against pathogens and supplying nutrients to other microbial residents within the gut ecosystem [70]. Recent studies have explored the relationship between physical activity and the presence of specific *Bacteroidetes* species, including *Prevotella intermedia*, *Bacteroides caccae*, and *Parabacteroides*. For instance, individuals engaging in high-intensity sports exhibit significantly higher levels of *Prevotella intermedia* and *Bacteroides caccae* [37]. Additionally, *Parabacteroides* are found in higher abundance in physically active adults [71,72]. A recent systematic review also highlighted that sustained exercise routines are associated with notable increases in *Prevotella* levels [73]. This relationship underscores the dynamic nature of *Bacteroidetes*, which can adapt and shift in response to exercise duration and intensity.

### 4.4. The Role of the Verrucomicrobia Phylum in Gut Health

Several studies reported a specific increase in the species *Akkermansia muciniphila* of the *Verrucomicrobia* phylum [54,61,67,74]. *Akkermansia muciniphila* has been shown to play a protective role in preventing various metabolic disorders, including diabetes. [61]. Zhong et al. studied the effects of aerobic and resistance exercise, finding a rise in SCFA-producing bacteria linked with anti-inflammatory effects such as *Verrucomicrobia* [75].

### 4.5. The Role of the Actinobacteria Phylum in Gut Health

*Actinobacteria* are one of the four main phyla of the human gut microbiome, and, despite their tiny number, they play an important role in gut homeostasis [76] *Bifidobacterium*, a genus of the *Actinobacteria* phylum, stands out due to its prevalence in physically active adults [54,62,64,74,77,78,79]. Studies have shown that engaging in high-intensity activity can lead to significantly higher levels of *Bifidobacterium* [48].

*Bifidobacterium* produces acetate, a short-chain fatty acid (SCFA) that protects intestinal epithelial cells from apoptosis triggered by the O157 toxin of *Escherichia coli* [80]. Acetate also plays a crucial role in inducing goblet cell differentiation, mucin secretion, and the sialylation process, all of which are essential for maintaining a healthy and functional gut barrier [80,81]. *Bifidobacterium* also produces lactate, which further stimulates through cross-feeding the production of other beneficial SCFAs such as butyrates by *Lactobacilli*, *Bacteroides*, or certain *Enterobacteria* [82].

### 4.6. The Role of the Proteobacteria Phylum in Gut Health

*Proteobacteria*, which include species such as *Escherichia coli*, play a critical role in indicating disturbances within the gut microbiome [83]. An imbalance characterized by elevated levels of Proteobacteria is often linked to poorer gut health and can exacerbate conditions such as chronic intestinal inflammation in mice [84]. Many studies have shown that the abundance of *Proteobacteria* decreases with moderate physical activity [38,48,75,78,85,86]. Zhong et al., who studied the effects of aerobic and resistance exercise, also found a decrease in pro-inflammatory bacteria such as *Proteobacteria* [75]. However, elite athletes participating in intense exercise showed a modest rise in *Proteobacteria*, which might be attributed to adverse effects on the gut microbiota caused by overloaded training [52,54,87]. Tabone et al., who studied fecal samples from athletes who completed a treadmill exercise, found an increase in *Proteobacteria*’s *Escherichia coli* species. [50].

## 5. Probiotic Supplementation in Sports Performance

Probiotic supplementation has garnered significant attention in sports science for its potential to enhance athletic performance through improved gut health and various physical parameters (Appendix A; Figure 2). Certain sports supplements like whey protein, polyphenols, and other compounds have also been shown to interact with the gut microbiota by promoting the growth of healthy species and limiting species producing toxic metabolites [88]. Creatine, a popular dietary supplement among athletes, has been shown to increase *Bacteroides* and *Firmicutes* species while decreasing *Proteobacteria*, *Fusobacteriota*, *Crenobacter*, and *Shewanella* species in animal models [89]. Intense exercise can lead to injuries, often requiring antibiotic treatment. Antibiotics, while necessary for treating infections, can disrupt the gut microbiome [90], potentially counteracting the benefits of probiotic supplementation. This dual impact on the gut highlights the need for a more comprehensive approach to managing gut health in athletes, particularly those undergoing bacterial infection treatment.

A blend of *Bifidobacterium* and *Lactobacillus* strains at high doses experienced increased VO2 max and extended exercise durations before failure, alongside reduced heart rates during activity [91]. Elite cyclists taking a probiotic mixture of *Bifidobacterium animalis* and *Lactobacillus helveticus* experienced significantly less nausea, vomiting, and inflammation during training [92]. Male runners on a 45 billion CFU multi-strain probiotic regimen for four weeks demonstrated extended running times before fatigue under heat conditions [93]. Elite rugby players consuming a diverse probiotic regimen (including *Lactobacillus acidophilus*, *Bifidobacterium animalis*, *Bifidobacterium bifidum*, *Bifidobacterium lactis*, *Streptococcus thermophilus,* and *Saccharomyces boulardii*) reported alleviated leg heaviness and muscle pain, which likely contributed to improved performance and recovery [94].

A study was conducted on trained volunteer athletes who were given a combination of probiotics including *Bifidobacterium bifidum*, *Bifidobacterium lactis*, *Enterococcus faecium*, *Lactobacillus acidophilus*, *Lactobacillus brevis*, and *Lactococcus lactis* at a dose of 1 × 10¹⁰ CFU/day for 12 weeks. The researchers found an increase in the ratio of subjects in the control group who had one or more upper respiratory tract infection (URTI) symptoms, suggesting that the intervention group had a better immune response and potentially better overall health and endurance [95]. Elite athletes administered *Lactobacillus helveticus,* which significantly reduced the duration and severity of upper respiratory tract infection (URTI) symptoms and led to them experiencing an increased sense of energy [96]. Division I female athletes supplementing with *Bacillus subtilis* reported improvements in strength measures and reductions in body fat percentage, indicating enhanced physical performance and recovery capabilities [97]. Badminton players consuming Lactobacillus casei with orange juice daily showed improvements in aerobic capacity [98]. Athletes supplemented with Lactococcus lactis exhibited increased CD86 expression and decreased cumulative days of fatigue, indicating improved immune function and reduced fatigue, which is particularly beneficial for endurance sports [99].

Marathon runners taking probiotics containing *B. animalis*, *B. bifidum*, and *L. acidophilus* for 28 days reported fewer gastrointestinal symptoms and enhanced gut health during strenuous exercise [100]. Probiotics may mitigate exercise-induced oxidative damage and inflammation, which are integral to maintaining athletic performance. Endurance-trained men taking multiple probiotics containing *B. bifidum*, *B. lactis*, *E. faecium*, *L. acidophilus*, *L. brevis*, and *Lactococcus lactis* exhibited reduced markers of inflammation and oxidative stress, along with decreased fecal zonulin levels, suggesting improved intestinal barrier function [101]. Athletes consuming *Lactobacillus paracasei* and *Lactobacillus rhamnosus* showed increased plasma antioxidant levels and passive reactive oxygen species (ROS) [102]. Triathletes administered *Lactobacillus plantarum* demonstrated significant reductions in inflammatory markers such as TNF-α, interleukin-6 (IL-6), and interleukin-8, along with increased levels of anti-inflammatory IL-10 and plasma branched-chain amino acids [103]. By minimizing inflammation, probiotics enable athletes to maintain higher levels of performance and comfort during training and competitions.

Probiotic supplementation can alter the intestinal flora, alleviating gastrointestinal issues and enhancing athletic performance as discussed in an earlier section. It also reduces fatigue signs and muscular soreness, which indirectly contributes to athletic performance. It also reduces the immune-suppressive effects and upper respiratory tract infections induced by vigorous physical activity as discussed. Probiotics can decrease the expression of nuclear factor kappa β (NF-κB) and pro-inflammatory cytokines while increasing levels of anti-inflammatory cytokines [104]. Anti-inflammatory cytokines help to reduce muscle inflammation and degeneration. Probiotics also influence the function of macrophages, which are crucial for muscle healing [105]. During physical exercise, muscle fibers generate myokines such as anti-inflammatory IL-10 and IL-6, which also possess systemic anti-inflammatory properties [106].

Oxidative stress and exercise-induced hyperthermia can lead to gastrointestinal issues during endurance exercise. Probiotics can help maintain intestinal integrity by enhancing the uptake of glucose and amino acids during extended physical activity. After physical activity, probiotics that include species such as *L. paracasei* can enhance the absorption of branched-chain amino acids, which are essential for muscle growth and repair [107]. Probiotics containing *L. plantarum* may improve iron utilization, which boosts aerobic capacity and endurance by accelerating erythropoiesis [108]. Additionally, *L. plantarum* supplementation can enhance skeletal muscle glycogen storage, providing a readily available energy source during physical activity and increasing endurance [109]. However, research on the potential of probiotics in nutritional metabolism related to exercise is still limited.

## 6. Potential Mechanisms of Action

The bidirectional association between physical activity and gut microbiota presents several potential implications for general health and sports nutrition. Exercise can improve intestinal health and metabolism, while probiotics can impact muscle function and athletic performance (Figure 3). This section delves into the latest research to explore two main areas: the potential mechanisms by which physical activity influences gut health and how the gut microbiome affects muscle metabolism and overall health.

### 6.1. Potential Mechanisms of Physical Activity in Gut Health

The link between exercise and the gut microbiota is multifaceted and bidirectional, with considerable effects on metabolic and muscle functioning, supporting the idea of a gut–muscle axis. This section will explore the mechanisms of how physical activity can influence the gut microbiota (Figure 3).

Exercise can impact gut health through several interconnected mechanisms. During exercise, mitochondria in skeletal muscles produce reactive oxygen and nitrogen species (RONS) [110]. RONS can trigger immune responses in the cells lining the colon through serotonin signaling pathways [111]. The effectiveness of the immune response is closely linked with the development of the gut microbiota [112]. Moderate aerobic exercise boosts the production of immunoglobulin A in the gastrointestinal tract, enhancing the gut microbiota’s ability to prevent intestinal pathogen colonization [113]. Regular moderate-intensity exercise, particularly resistance training, reduces circulating levels of lipopolysaccharides (LPS) and decreases the expression of Toll-like receptors [114]. This reduction is associated with decreased inflammation and improved gut barrier function [115].

Exercise has been shown in studies to have an essential role in controlling bile acid pools, which promotes the health of the host gut microbiota [116]. Bile salt metabolism is an essential component of the gut microbiota. The secondary bile acids generated by the microbiota are detected by the tissues via the activation of the Farnesoid X receptor (FXR) and the G-protein-coupled bile acid receptor, which play critical roles in energy metabolism. The gut microbiota can efficiently control the body’s metabolic capacity and muscle anabolism by inhibiting FXR [117]. Hydrogen sulfide is another messenger generated by cysteine breakdown by gut microbiota that controls the metabolism of intestinal epithelial and immunological cells [118]. Hydrogen sulfide may be a possible target for enhancing myogenesis and improving muscle function. Supplementation with sodium hydrosulfide stimulates myogenic gene expression and improves the regeneration of muscles in mice [119].

### 6.2. Potential Mechanisms of the Gut Microbiome in Muscle Metabolism and Health

The gut microbiome regulates muscle metabolism via pathways involving AMP-activated protein kinase (AMPK) and downstream signaling pathways such as the mammalian target of rapamycin (mTOR). These pathways control a variety of physiological activities like glucose, protein, and lipid synthesis [120]. Studies suggest that SCFAs can activate muscle AMPK directly by raising the adenosine monophosphate/triphosphate (AMP/ATP) ratio [121].

The gut microbiome significantly influences muscle metabolism through critical pathways such as AMP-activated protein kinase (AMPK) and the mammalian target of rapamycin (mTOR). Insulin Growth Factor (IGF-1), a primary modulator of bone and muscle development, has a considerable impact on the mTOR pathway, which is critical for muscle synthesis [122]. These pathways thus are pivotal in regulating essential physiological functions, including glucose, protein, and lipid synthesis [120]. Physical exercise benefits from this synthesis as it enhances energy production and metabolic efficiency.

A strong correlation exists between SCFAs, produced by gut bacteria, and muscular strength [121,123]. SCFAs can directly activate AMPK in muscles [121]. This activation is beneficial during physical exercise as it plays an important role in muscle development and growth [124]. Mice treated with antibiotics to disturb the gut microbiome demonstrated reduced soleus and plantaris hypertrophy after wheel running as compared to those not treated [125]. One notable SCFA, butyrate, promotes the release of glucagon-like peptide-1 (GLP-1) from colonic cultures, suggesting that it could play a role in reducing muscle atrophy [126]. Butyrate specifically supports aerobic metabolism in skeletal muscles by boosting mitochondrial activities such as increasing oxidative phosphorylation, further underscoring the importance of SCFAs in muscle health and performance [127].

The gut microbiota plays a significant role in protein metabolism. In the small intestine, bacteria such as *Clostridium*, *Bacillus*, *Streptococcus*, and *Proteobacteria* are among the most abundant species involved in amino acid fermentation, while *Clostridia* is particularly prevalent in the large intestine [128]. Additionally, these bacteria can enhance the host’s ability to absorb amino acids. For instance, *Bacillus coagulans* can increase amino acid levels in the serum following milk protein consumption [129].

The gut microbiome can affect the production of hormones such as insulin and androgens, which are essential for protein synthesis and muscle development [130]. Minimal or excessive physical activity can lead to a disturbed gut microbiome, which can have profound effects on these processes. For instance, the bacterium *Bacteroides*, known for its pro-inflammatory properties, promotes insulin resistance by enhancing polysaccharide fermentation, lipogenesis, and the development of host adipose tissue [40]. Elevated levels of *Bacteroides* are commonly observed in individuals with hyperandrogenic conditions, such as Polycystic Ovary Syndrome [131]. Recent studies have shown that androgen insufficiency, induced by castration, not only alters the gut flora but also increases the risk of obesity and the loss of thigh muscle in mice [132]. Interestingly, the use of antibiotics has been found to minimize these alterations [132].

Recent research has emphasized the gut–brain–muscle axis, which connects a healthy gut with athletic performance. Research in animal models suggests that the gut microbiota may influence mood and stress by interacting with the central nervous system. In mice, it has been found that intestinal microbiota’s fatty acids stimulate afferent sensory neurons, which signal the brain to suppress monoamine oxidase (MAO) expression during exercise. This suppression leads to increased dopamine signaling, which in turn motivates exercise behaviors and enhances performance. [133].

Stress initiated by exercise can stimulate the hypothalamic–pituitary–adrenal axis, resulting in the production of cortisol and noradrenaline, both of which affect the gut microbiota [134]. The production of corticotropin-releasing factor also alters gastrointestinal function, influencing inflammatory processes, colonic transit duration, mucosal secretory functions, barrier functions, and the growth of intestinal tract bacteria [135]. Additionally, stress can increase plasma noradrenaline levels, affecting the gut microbiota and increasing the aggressiveness of infectious enteric bacteria [136]. Excessive physical activity can raise body temperature and cause extreme heat stress, negatively impacting the gastrointestinal microbiome. It reduces gut blood flow, resulting in relative ischemia and increased intestinal permeability, which may allow bacteria to migrate from the intestines and cause gastrointestinal issues [137].

## 7. Future Research and Perspectives

While probiotics have shown beneficial effects for athletes, our analysis identified several limitations in the studies that could potentially bias our analyses. Research into the modulation of the gut microbiome through probiotics has utilized a variety of strains, dosages, and supplementation durations, complicating direct comparisons and the ability to draw definitive conclusions about the most effective strains or dosages. Furthermore, the lack of standardized outcomes across these studies adds another layer of complexity. While some investigations focus on aerobic capacity and VO_2_ max, others might assess muscle soreness or gastrointestinal symptoms. This inconsistency in measured outcomes hampers the establishment of a consistent set of metrics for evaluating the efficacy of probiotic supplementation. Additionally, variations in demographic characteristics, such as age, sex, and baseline health status, across studies could influence outcomes and their applicability. Notably, most studies administering probiotics have been conducted with male athletes, with fewer focusing on female athletes, potentially affecting the interpretations due to physiological differences between genders.

Current research predominantly focuses on bacteria within the gut microbiota, yet this complex ecosystem comprises a broader array of microbial taxa, including fungi and archaea. To fully understand the interactions between physical activity and gut health, future research should expand its scope to assess these diverse microbial communities. This approach will help to identify the most beneficial types of exercise prescriptions tailored to individual microbiome profiles. Additionally, reporting microbiome data at the species level, rather than at the broader phylum level, would provide more precise insights that are crucial for developing targeted clinical interventions. This level of detail is essential for translating microbiome research into practical health solutions.

The impact of physical activity on the gut microbiome remains uncertain and varies substantially across studies. To enhance reliability and precision, standardized study designs such as RCTs with larger sample sizes are necessary. Many current cross-sectional studies rely on self-reported questionnaires that might include biases and inaccuracies that affect the research results. More comparable results regarding the effects of physical activity on the gut microbiome could be achieved by employing more objective measurement techniques like wearable fitness trackers. Rather than measuring vastly different outcomes, there should be a core set of metrics that all studies should include, such as microbial diversity changes and the intensity of exercise, to allow for better cross-study comparisons. These studies should also consider potential confounding factors such as BMI, nutrition, and biotic consumption to better isolate the specific effects of physical activity. A broader range of environmental factors, such as geographic location, pollution, and antibiotic use can also profoundly impact gut health. By controlling for these variables, researchers can better isolate the effects of probiotics or physical activity on the microbiome. Future multi-omics studies could provide a holistic view of the biological effects of physical activity, which will help to unravel the numerous genes and molecular pathways that are altered.

## 8. Conclusions

Research suggests that the impact of physical activity on gut microbiota differs based on the type, intensity, and duration of exercise. Regular exercise enhances the growth of beneficial bacteria and short-chain fatty acids (SCFAs), improving gut health and potentially warding off metabolic diseases. Furthermore, exercise boosts microbial diversity, which is vital for both metabolic and immune health, and enhances an anti-inflammatory state throughout the body. The alteration of the beneficial gut microbiota with exercise and probiotics increases compounds including SCFAs, bile acids, and hydrogen sulfide, which can improve muscle mass and exercise endurance. Furthermore, research indicates that gut flora may play a crucial role in macronutrient absorption and modulate neuroendocrine pathways, resulting in enhanced gut health, increased athletic ability, and fewer health issues. More research on metabolic and physiologic pathways is needed to better describe the relationship between physical activity and gut microbiota.

## Figures and Tables

**Figure 1 nutrients-16-03663-f001:**
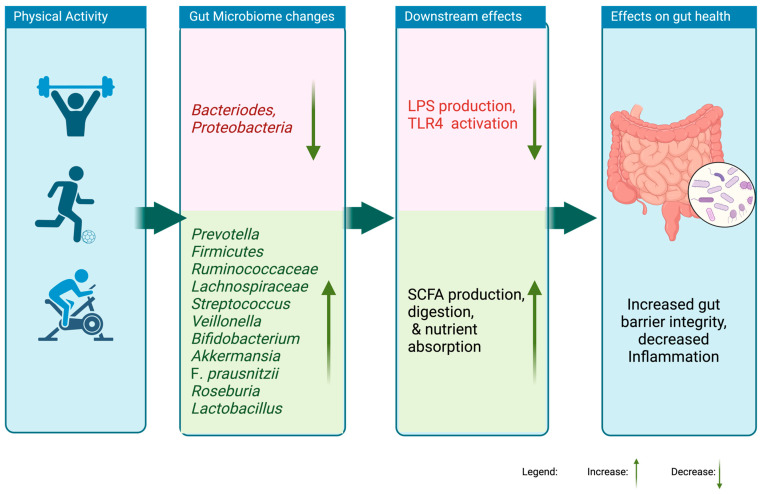
Effects of physical-activity-induced changes in the gastrointestinal microbiome.

**Figure 2 nutrients-16-03663-f002:**
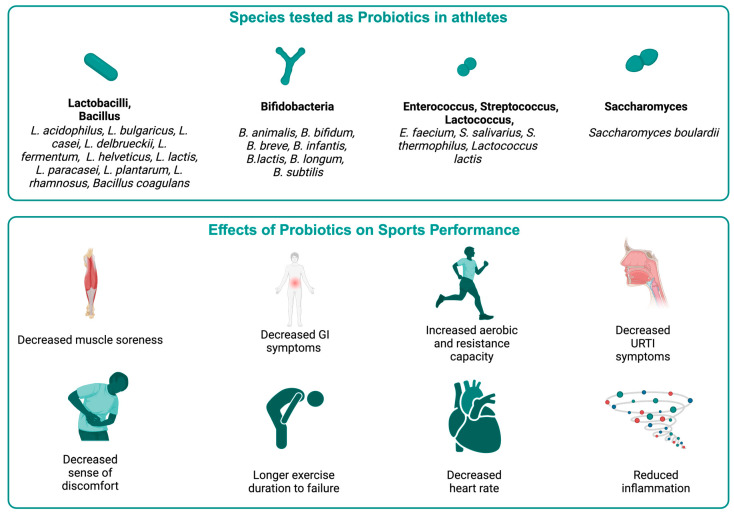
Effects of probiotics on athletic health and performance.

**Figure 3 nutrients-16-03663-f003:**
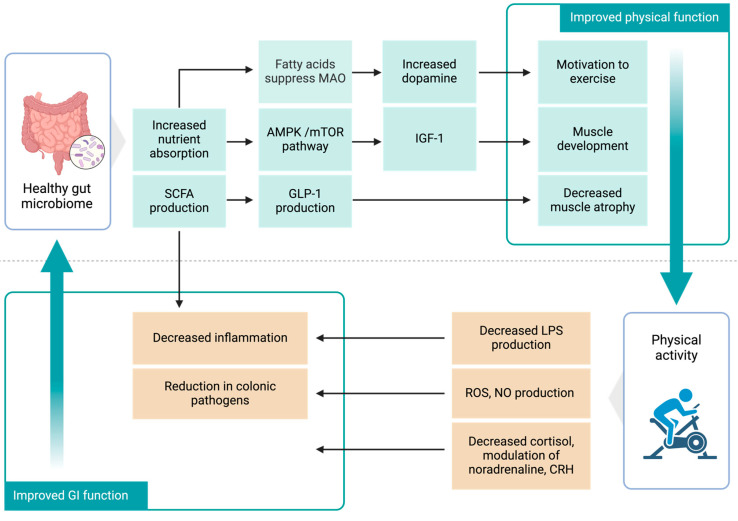
Mechanisms underlying the mutual relationship between physical activity and the gut microbiome.

## Data Availability

No new data were created in this study. The data that support the findings of this study are included within the article and Appendix A.

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
