# Peer review of "Physical Exercise and the Gut Microbiome: A Bidirectional Relationship Influencing Health and Performance"

_nutrients, 2024, doi:10.3390/nu16213663_

Round 1

Reviewer 1 Report

Comments and Suggestions for Authors

The topic of this review is fascinating, as it explores the connection between physical exercise and microbiota. The authors presented studies that examine how different forms of exercise may influence the composition and diversity of the gut microbiome. However, I suggest improvements in several areas of this text, particularly regarding the general review and content. Some subjects are addressed superficially, and it would be beneficial for the authors to enrich these sections with more in-depth information about the differences in microbiota and related physiological mechanisms. A more comprehensive discussion could provide valuable insights into how various factors influence microbiota diversity and function and the implications for health and disease. This additional depth would enhance the overall quality of the review and better inform readers. Additionally, understanding the interplay between exercise and microbiota may lead to new therapeutic approaches to promoting health and preventing diseases.

As this is a review study, it is crucial for the authors to elaborate on the types of exercise discussed. Please include more details regarding the studies that compare exercise type, intensity, and duration. It’s important to make this section as informative as possible for readers.

-       Please revise the entire text to ensure all bacterial names are italicized. Additionally, check for grammar and formatting issues.

Line 29: add more descriptive information: “the human microbiome is a complex community of microorganisms, including bacteria, archaea, fungi, viruses, and protozoa.

Line 31, 34: add “anaerobic" species”

Line 3": add antibiotics

Line 41: remove facultative anaerobe and keep “facultative species” only.

Line 49: Please revise this statement because Bacteroides is not considered pathogenic. In general, Bacteroides spp. are considered commensal bacteria rather than pathogenic. However, they can become opportunistic pathogens under certain circumstances, such as when there is a disruption to the normal gut barrier. 

Line 96: What do the authors consider voluntary aerobic exercise? Please provide more details regarding the activities and relevant studies, as this section appears incomplete."

Line 248: Lactococcus doesn’t belong to the Bacteroidetes species; they are part of the Firmicutes.

Line 256: Where is the increase in Akkermansia? Also, are some studies correlating it with the improvement of mucus in the intestine during exercise?

Line 279: Please clarify the intensity or type of exercise. Here, the authors state that the abundance of Proteobacteria decreases during exercise; however, in another part of the text, it was mentioned that intense exercise increases Proteobacteria. 

Line 286: Besides probiotics and prebiotics, the authors should also briefly comment on sports supplementation. Do whey protein, creatine, and pre-and post-workout recovery supplements influence this balance or dysbiosis? 

Line 430 – 451: Please move this paragraph to the probiotics section.

Comments on the Quality of English Language

N/A

Reviewer 2 Report

Comments and Suggestions for Authors

This is a well-written review paper about the relationship between physical exercise and gut microbiome. The background and the summary of the recent research in this area are sound and comprehensive. I believe this review will get a lot of attention from the relative audience.  

The only minor comment is to add a suitable reference for the sentence in lines 46-47. 

Reviewer 3 Report

Comments and Suggestions for Authors

The paper presents a narrative review exploring the bidirectional relationship between physical exercise and the gut microbiome, with a focus on the potential benefits of probiotic supplementation for sports performance. While the topic is relevant and timely, given the increasing interest in gut health and its connection to overall well-being, there are several points to be addressed:

Abstract:

The abstract mentions a "bidirectional relationship" between physical activity and the gut microbiome but does not clearly explain how this mutual influence occurs.

The abstract states that over 100 studies were reviewed, but it does not describe how the studies were selected or assessed for quality.

The abstract touches on the positive effects of probiotics but does not specify which strains were studied or how they impacted gut health or sports performance. Including more details about the strains used, dosage, and duration of supplementation would improve the understanding of the benefits.

The conclusion could be stronger by summarizing the most significant findings and their potential practical applications. This would provide a clearer takeaway for readers.

Introduction

The introduction mentions important concepts such as SCFA production, microbial imbalances, and the effects of exercise, but it fails to provide enough specificity regarding the underlying mechanisms. For instance, the production of SCFAs by anaerobes is crucial, but more detail could be provided about how these metabolites interact with the host’s immune system and gut integrity. Similarly, the effects of microbial diversity on metabolic processes are only briefly mentioned, and greater depth would strengthen the narrative.

The transition between discussing gut microbiome dysbiosis and the role of physical activity feels abrupt. The two topics are central to the review, yet they are not linked smoothly in the narrative. The introduction should better connect how physical activity specifically influences gut health and SCFA production beyond general immune and metabolic benefits.

While the introduction cites various studies linking physical activity to improvements in gut health, many of these are likely correlations rather than causations. The introduction should acknowledge that the exact causal mechanisms behind these correlations are still under investigation.

The final sentence states that the review aims to identify gaps in the current literature, but the introduction does not specify what these gaps might be. A stronger setup of these research gaps would make the article’s objectives clearer and give the reader a better sense of what to expect.

Chapter 4

Discussion of interconnection between exercise and energy intake/diet would broaden the impact of the review. A physical activity-induced hunger and the effect of energy dense diet on gut microbiome (https://www.nature.com/articles/s41387-020-0119-4 ) should be discussed.

Chapter 5

Probiotic supplementation has gained significant attention in sports medicine for its potential to enhance athletic performance by improving gut health. However, it is important to also consider that intense exercise can lead to injuries, often requiring antibiotic treatment. Antibiotics, while necessary for treating infections, can disrupt the gut microbiome (https://pubmed.ncbi.nlm.nih.gov/37760733/ ), potentially counteracting the benefits of probiotic supplementation. This dual impact on the gut highlights the need for a more comprehensive approach to managing gut health in athletes, particularly those undergoing injury treatment. Discussion of the above interaction would strengthen the review impact.

Future Research and Perspectives

The chapter provides an insightful reflection on the limitations of current research and offers valuable recommendations for future studies. However, it could benefit from more specificity in terms of standardizing research methodologies, and a broader consideration of environmental factors that influence gut health. By addressing these gaps, the chapter would present a more comprehensive and actionable framework for future research in the field.

Comments on the Quality of English Language

Minor edits to improve flow are recommended.

Round 2

Reviewer 1 Report

Comments and Suggestions for Authors

Line 324- 327: Please remove this statement, as antibiotics are not used to treat injuries. We must be careful, as antibiotics are only recommended for bacterial infections. Given the widespread issue of antimicrobial resistance, we must be cautious about what we suggest or write in our papers.

Author Response

Comments 1: Line 324- 327: Please remove this statement, as antibiotics are not used to treat injuries. We must be careful, as antibiotics are only recommended for bacterial infections. Given the widespread issue of antimicrobial resistance, we must be cautious about what we suggest or write in our papers.

Response 1: Thank you for pointing that out. We have removed it accordingly.